# Hyperspectral Bare Soil Index (HBSI): Mapping Soil Using an Ensemble of Spectral Indices in Machine Learning Environment

**Eric Ariel L. Salas** * and **Sakthi Subburayalu Kumaran**

Agricultural Research Development Program (ARDP), Central State University, Wilberforce, OH 45384, USA; ssubburayalu@centralstate.edu
* Correspondence: esalas@centralstate.edu

**Abstract:** Spectral remote-sensing indices based on visible, NIR, and SWIR wavelengths are useful in predicting spatial patterns of bare soil. However, identifying an effective combination of informative wavelengths or spectral indices for mapping bare soil in a complex urban/agricultural region is still a challenge. In this study, we developed a new bare-soil index, the Hyperspectral Bare Soil Index (HBSI), to improve the accuracy of bare-soil remote-sensing mapping. We tested the HBSI using the high-spectral-resolution AVIRIS-NG and Sentinel-2 multispectral images. We applied an ensemble modeling approach, consisting of random forest (RF) and support vector machine (SVM), to classify bare soil. We found that the HBSI outperformed other existing bare-soil indices with over 91% accuracy for Sentinel-2 and AVIRIS-NG. Furthermore, the combination of the HBSI and the normalized difference vegetation index (NDVI) showed a better performance in bare-soil classification, with >92% accuracy for Sentinel-2 and >97% accuracy for AVIRIS-NG images. Also, the RF-SVM ensemble surpassed the performance of the individual models. The novelty of HBSI is due to its development, since it utilizes the blue band in addition to the NIR and SWIR2 bands from the high-spectral-resolution data from AVIRIS-NG to improve the accuracy of bare-soil mapping.

**Keywords:** bare-soil index; hyperspectral bare soil index; soil mapping; urban–agricultural complex



## 1. Introduction

Bare soil is crucial for an understanding of the ecosystem structure, and has become a key driver of ecological functioning [1]. The dynamics of bare soil is a focus in soil evaporation studies and in the quantification of the overall water balance [2,3], the prediction of dust deposition [4], and the assessment of urban development [5]. Bare-soil features, such as albedo, are widely used in climate models for the retrieval of satellite-based land surface properties [6]. Given the importance of bare soil, there is a need to enhance current predictive methods and use publicly available spatial data to produce bare-soil dynamic mapping estimations.

Remote sensing (RS) data have long been documented as an important variable in predictive soil mapping [7]. For this purpose, soil indices (SI) derived from RS images have been consistently utilized [7]. However, misclassification of bare soil often occurs due to the challenges in accurately capturing the diverse spectral characteristics of bare-soil parcels using these soil indices. SIs are specifically designed to detect bare-soil parcels using various wavelength bands that are believed to be sensitive to bare soil, including visible (VIS) to near-infrared (NIR: 750–850 nm) and shortwave infrared (SWIR: 900–2500 nm) [8,9]. The use of VIS-NIR-SWIR for the spectral analysis of soil characteristics (e.g., soil organic carbon, pH, bulk texture) has grown in popularity over the past decades [10]. The bare-soil index (BSI), for example, was formed using the SWIR2 (~2080–2350 nm) and green band (500–600 nm), and is used to map potential areas of soil degradation and erosion [11]. BSI, in combination with the normalized difference vegetation index (NDVI), also discriminates exposed soil from vegetation cover when applied to an intensive agricultural area [12].

The bareness index (BI), on the other hand, utilizes the red (600–700 nm), NIR, and SWIR1 (~1570–1650 nm) to detect non-vegetated and urban area classes [13]. Other soil indices only used a combination of red and NIR regions [14] or red and thermal infrared (TIR: 10,000 to 11,200 nm) [15] for the detection of barren soil. Overall, these SIs have been widely utilized in land use-landcover (LULC) studies, specifically to distinguish bare land features from urban classes. However, in LULC classification, bare soil is frequently misclassified as an urban area, and vice versa, because of their overlapping spectral characteristics, and this can lead to challenges in accurate differentiation [16]. This misclassification issue has significantly hindered the effective mapping of urban areas using remotely sensed data, underscoring the need for extensive research and innovative approaches. To address this challenge, it is important to develop algorithms that can effectively identify and highlight the spectral bands that are proven to be the most useful for detecting bare soil. These algorithms must be capable of differentiating between bare soil and urban areas even in complex urban/agricultural regions, where the spectral signatures may vary due to diverse landcover compositions and environmental conditions.

Several studies have been published that used VIS-NIR and SWIR for soil characterization [17]. These wavelengths are known to provide valuable information about the reflectance properties of soil, allowing researchers to extract meaningful insights related to soil composition, moisture content, and other key parameters [17,18]. However, few attempts have been made to identify effective combinations of spectral bands and indices for mapping bare soil in a complex urban/agricultural region. By exploring novel combinations of spectral bands and indices, and developing advanced analytical techniques, we could enhance the precision and reliability of bare-soil mapping in urban/agricultural regions. Therefore, the major goal of this study was to identify and map bare-soil patches in an urban/agricultural site located in Anand, India using a new bare-soil index. To that aim, we tested the new soil index, compared it with existing soil indices, and used machine learning algorithms to classify two types of images: multispectral and hyperspectral images. We used publicly accessible high-spectral-resolution AVIRIS-NG and multispectral Sentinel-2 multispectral images as a low-cost and efficient method of mapping bare soil. We put a special emphasis on the dependability of the new bare-soil index that we developed, as well as on the combination of spectral indices for successful bare-soil identification and mapping.

## 2. Materials and Methods

### 2.1. Study Area

We selected the Anand District in the state of Gujarat, India as the study area (Figure 1). The site covers approximately 216 square kilometers (21,600 ha). The district falls within the Western Plain and Hill agroecological sub-region (ICAR). The average altitude of this region is 43 m above mean sea-level (MSL). The soil types in this region fall into the following two main categories: sandy loam and clay loam soils. The normal rainfall in this region is 687 mm, generated by the southwest monsoon (June–September). The major crops grown in the district include kharif-season irrigated rice, rabi-season irrigated and rainfed wheat, and kharif-season irrigated pearl millet, tobacco, cotton, and vegetables (potato, brinjal, tomato, and cabbage). The investigated site represents farmlands with diverse agricultural management and land-use systems.

### 2.2. Remote Sensing Dataset

We used available hyperspectral airborne visible infrared-imaging spectrometer-next-generation (AVIRIS-NG) imagery with a nominal ground resolution of 4 m to map the bare soil, taken on 26 March 2018. AVIRIS-NG samples 430 contiguous bands between 380 nm and 2510 nm at approximately 5 nm spectral resolution. An ortho-corrected and atmospherically corrected reflectance dataset (L2) for the study area is archived through NASA (https://aviris-ng.jpl.nasa.gov, accessed on 20 November 2022). We also used Sentinel-2 multispectral images available from the European Space Agency (ESA) Sentinels Scientific

Data Hub (https://scihub.copernicus.eu/, accessed on 20 November 2022). Sentinel-2 has 13 spectral bands: 4 bands at 10 m resolution, 6 bands at 20 m resolution and 3 bands at 60 m spatial resolution. The orbital swath width is 290 km. In this work, we downloaded the associated Sentinel-2 level 2A scenes taken on 20 March 2018 that were cloud-free and available for all sampling locations. Distributed level 2 products were atmospherically corrected by the Sen2Cor package (https://step.esa.int/main/snap-supported-plugins/sen2cor/, accessed on 20 November 2022). Both images matched up with, and were within, the field campaign days. We resampled all bands that were in 20 m resolution to 10 m to be consistent with the four native bands (band 2 in blue, band 3 in green, band 4 in red, and band 8 in NIR).

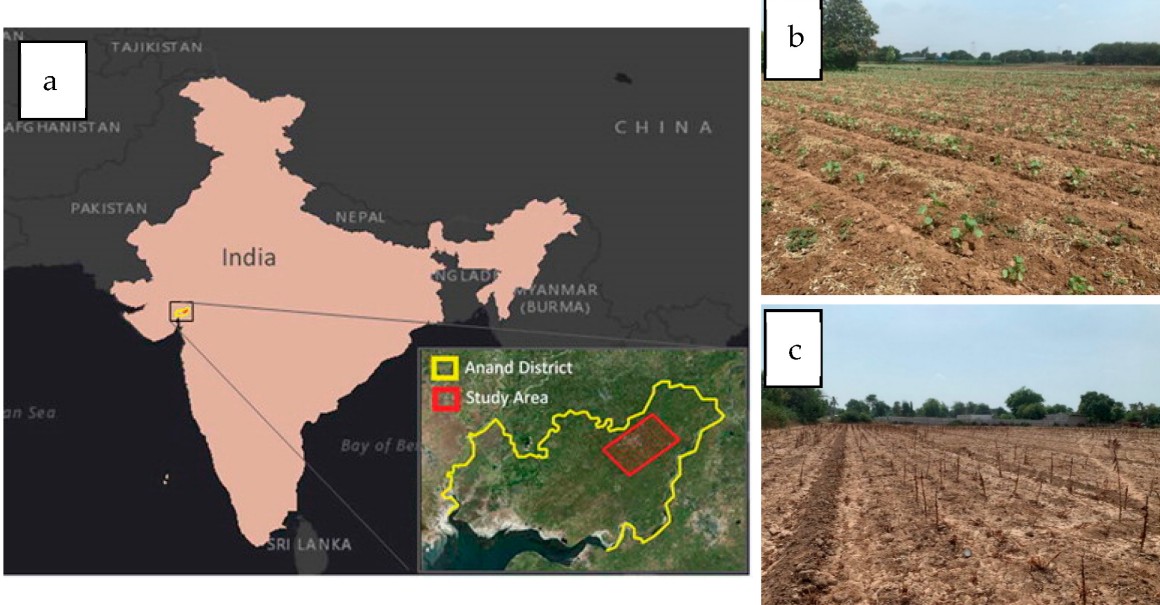

**Figure 1.** (**a**) Map showing the location of the study site in the Anand District in Gujarat, India. Examples of farmlands with sparse vegetation and bare-soil conditions are shown in (**b**,**c**), respectively.

### 2.3. Image Indices

We utilized RS image indices to differentiate land features (Table 1). Six vegetation indices (VI) including the normalized difference vegetation index (NDVI) [19], modified soil adjusted vegetation index 2 (MSAVI2) [20], soil adjusted vegetation index (SAVI) [21], enhanced vegetation index (EVI) [22], transformed vegetation index (TVI) [23], and the green normalized difference vegetation index (GNDVI) [24] derived from the AVIRIS-NG and Sentinel-2 images. Combining VIs and SIs to map bare soil minimizes vegetation influence and maximizes bare-soil features in order to extract the pixels most likely to be bare. Also, adding VIs could leverage the synergistic relationships between vegetation and soil properties, thereby improving the accuracy and robustness of the bare-soil mapping models. Agone and Bhamare [25] linked NDVI values of 0.00 to 0.20 with bare areas and several authors have used these VIs to classify bare-soil classes [26–29].

To verify the performance of our proposed index, we added a set of SIs to differentiate between bare soil, vegetation, and urban areas, including the bare soil index (BSI) [11]), brightness index (BI) [30], and normalized difference soil index 2 (NDSI2) [31]. These SIs were constructed with wavelength combinations from VIS to SWIR regions with the goal of differentiating between bare-soil status and vegetation status.

**Table 1.** Existing and new spectral indices used as covariates to map bare soil. For broadband indices, spectral bands were averaged to represent NIR (750–850 nm), red (600–700 nm), green (500–600 nm), blue (400–500 nm), and SWIR2.

| Index | Equation |
|:---:|:---:|
| **Vegetation Indices (VI)** | |
| NDVI | $\frac{Red-NIR}{Red+NIR}$ |
| MSAVI2 | $\frac{2NIR+1-\sqrt{(2NIR+1)^2-8(NIR-Red)}}{2}$ |
| SAVI | $\frac{(NIR-Red)(1+L)}{NIR-Red+L}$ <br> *where L = 1* |
| EVI | $Green * \frac{(NIR-Red)}{NIR+C_1*Red-C_2*Blue+L}$ <br> *where $C_1$ = 6, $C_2$ = 7.5, L = 1* |
| TVI | $\sqrt{\frac{NIR-Red}{NIR+Red}}+0.5*100$ |
| GNDVI | $\frac{NIR-Green}{NIR+Green}$ |
| **Soil Indices (SI)** | |
| BSI | $\sqrt{\frac{SWIR2-Green}{SWIR2+Green}}*100$ |
| BI | $\sqrt{\frac{Red^2+Green^2}{2}}$ |
| NDSI2 | $\frac{SWIR2-Green}{SWIR2+Green}$ |
| HBSI | $\frac{(SWIR2+Green)-(NIR+Blue)}{(SWIR2+Green)+(NIR+Blue)}$ |

*2.4. Hyperspectral Bare Soil Index (HBSI)*

We formulated the HBSI in Equation (1) by utilizing the spectral bands in the VIS, NIR, and SWIR2 regions. The HBSI takes advantage of the absorption bands at 400 to 500 nm (blue and green), the minimal absorption region around 900 nm (NIR), and the weak absorption near 2200 and 2300 nm (SWIR2). The VIS-NIR absorption in soil is mainly due to electronic transitions of the main active components of Fe-oxide minerals that do not have full d-orbitals [32]. Goethite ($\alpha$-FeOOH) and hematite ($\alpha$-Fe$_2$O$_3$) are the most common Fe-oxide minerals that exhibit broad absorption bands in the VIS-NIR regions. In the SWIR2, clay minerals and soil organic matter often show in narrow absorption spectral features [33,34]. Soil minerals that exhibit strong spectral maxima and minima of the second derivative in the SWIR2 region are gibbsite (Gbs) at around 2265 nm to 2285 nm, illite (Ill) at around 2205 nm to 2280 nm, and calcite (Cal) at around 2342 nm to 2367 nm [35].

$$HBSI = \frac{(SWIR2 + Green) - (NIR + Blue)}{(SWIR2 + Green) + (NIR + Blue)} \tag{1}$$

The spectral bands were averaged to represent the regions of green (500–600 nm), blue (400–500 nm), NIR (750–850 nm), and SWIR2 (~2080–2350 nm). With the HBSI, the reflectance spectra are normalized to allow a quantitative comparison between absorption features, such as depth, depth position, area of the absorption, etc., and to nondestructively retrieve bare soil [36].

*2.5. Machine-Learning Classification Algorithms*

We used random forest (RF) and support vector machine (SVM) classification algorithms to map bare soil [37] based on their proven performance and ability to handle complex classification tasks. We compiled a number of codes in R to run both algorithms [38,39]. RF and SVM are popular classifiers for digital soil-mapping using remote-sensing data to effectively capture complex relationships between spectral information and landcover classes, making it suitable for bare-soil detection [40,41]. RF and SVM provide a way to

select important covariates based on changes in the prediction accuracy when variables are added or deleted from models.

RF is a non-parametric supervised classifier that uses classification and regression tree (CART) through bagging, where it randomly picks a set of features and creates a classifier with a bootstrapped sample of the training data to grow a tree [35]. With RF training data selection, it is possible for the same sample to be picked several times, whereas others may not be picked at all. Apart from RF being quite robust with highly collinear variables, the random selection process at each tree node causes a low correlation among the trees and avoids over-fitting [42]. SVM is also a non-parametric supervised classifier used for pattern recognition, classification, and regression analysis. SVM is robust when processing a small number of training samples, but efficient at producing accurate maps when applied as a classifier to satellite images, as reported by Mountrakis et al. [43].

*2.6. Covariates, Training, and Test Datasets*

To construct the RF and SVM algorithms for mapping bare soil, we used the image indices from AVIRIS-NG and Sentinel-2 as covariates. For each algorithm, we ran the classification algorithms using two sets of covariates: one set that contained all six VIs and another set with all four SIs (Table 1). We also ran a model ensemble that combined the RF and SVM algorithms. Afterwards, we ran the classification models using only the top three and top two important VI and SI variables from the previous step, respectively. We ran the models for the last time using only the two most important VI and SI covariates from the previous step.

To produce the bare-soil maps, we limited the class features to only three: bare soil, urban areas, and others (forest, vegetation, etc.). We extracted high-quality samples from the study area using Google Earth, and each class label was visually interpreted using spectral profiles and using our expert knowledge of the field sites. These samples served as regions of interest (ROI) for the RF and SVM classification process. Each ROI was assigned to a specific LULC class. A total of 540 polygon samples were extracted: 200 urban, 200 bare soil, and 140 others. We then split the samples into training and testing sets. We used a splitting criterion of 70–30, where 70% of the sample data were used for calibration and 30% for model validation.

To evaluate the effectiveness of the classification models, we generated a classification error matrix. We utilized conventional accuracy metrics, including overall accuracy (OA) and kappa statistics (KA) to quantify the performance of the RF and SVM models. These metrics provided a comprehensive assessment of the models' ability to accurately classify bare soil, urban areas, and other landcover categories. By employing these indicators, we aimed to thoroughly evaluate the performance of the RF and SVM models in accurately mapping bare soil in the study area. The utilization of established accuracy metrics allowed for a reliable assessment of the performance of the models, enabling us to validate their efficacy in producing accurate bare-soil maps.

## 3. Results

Figure 2 shows the final classified bare-soil maps derived using an ensemble of RF and SVM from (a) AVIRIS-NG (4-m spatial resolution) and (b) Sentinel-2 (10-m spatial resolution) images. The map using Sentinel-2 shows that more areas were classified as bare soil (149 km$^2$) compared to the classification map using AVIRIS-NG (124 km$^2$) images. The magnified section (Figure 2c) shows the delineation of the bare-soil class from urban areas. Furthermore, the black box highlights the edges of a patch of urban areas misclassified as bare soil.

Table 2 lists the top important variables according to the RF and SVM classification models for each set of variables and images. For Sentinel-2 and AVIRIS-NG, NDVI and TVI were predominant variables for RF and SVM, ranking mostly in the top two in six for the VI set. The HBSI and NDSI2 were the most important variables for the SI set. When the important VI and SI variables were combined, HBSI and NDVI ranked first. Between the

NDVI and HBSI, the latter was the variable with a higher classification importance in both the RF and SVM models and images.

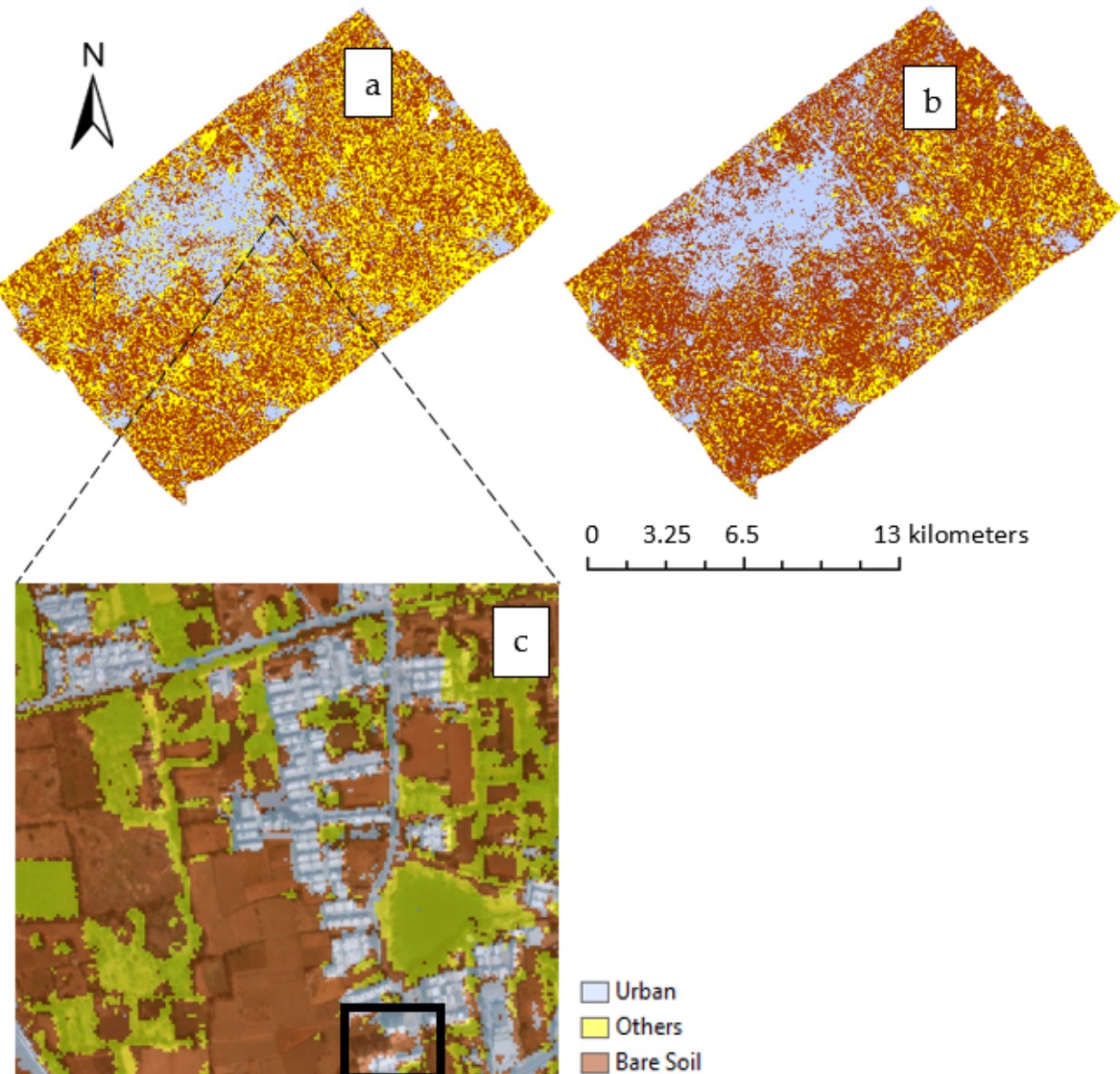

**Figure 2.** Bare soil-urban map derived using an ensemble of RF and SVM from: (**a**) AVIRIS-NG; (**b**) Sentinel-2 images; and (**c**) magnified section to highlight the separation of classes.

**Table 2.** Rankings of overall variable importance for RF and SVM models using different sets of covariates. The symbol * means that the variable does not have any contribution.

| Variable Set | Sentinel-2 | | AVIRIS-NG | |
|---|---|---|---|---|
| | RF | SVM | RF | SVM |
| 6 VIs | 1. NDVI | 1. TVI | 1. NDVI | 1. TVI |
| | 2. TVI | 2. NDVI | 2. GNDVI | 2. NDVI |
| | 3. GNDVI | 3. SAVI | 3. TVI | 3. GNDVI |
| | 4. EVI | 4. GNDVI | 4. SAVI | 4. SAVI |
| | 5. SAVI | 5. MSAVI | 5. EVI | 5. MSAVI |
| | 6. MSAVI | 6. EVI | 6. * MSAVI | 6. * EVI |

**Table 2.** *Cont.*

| Variable Set | Sentinel-2 | | AVIRIS-NG | |
| --- | --- | --- | --- | --- |
| | RF | SVM | RF | SVM |
| 4 SIs | 1. HBSI<br>2. NDSI2<br>3. BSI<br>4. BI | 1. HBSI<br>2. NDSI2<br>3. BSI<br>4. BI | 1. HBSI<br>2. NDSI2<br>3. BI<br>4. BSI | 1. HBSI<br>2. NDSI2<br>3. BSI<br>4. BI |
| 3 VIs &<br>2 SIs | 1. HBSI<br>2. NDVI<br>3. NDSI2<br>4. TVI<br>5. GNDVI | 1. NDVI<br>2. HBSI<br>3. NDSI2<br>4. GNDVI<br>5. TVI | 1. NDVI<br>2. HBSI<br>3. GNDVI<br>4. NDSI2<br>5. TVI | 1. HBSI<br>2. NDSI2<br>3. NDVI<br>4. TVI<br>5. GNDVI |
| NDVI & HBSI | 1. HBSI<br>2. NDVI | 1. HBSI<br>2. NDVI | 1. HBSI<br>2. NDVI | 1. HBSI<br>2. NDVI |

The Sentinel-2 overall validation accuracy from the final dataset with NDVI and HBSI were 93.5% (K = 87.8%), 94.9% (K = 90%), and 93.6% (K = 87.9%) for RF, SVM, and the ensemble, respectively (Table 3). For AVIRIS-NG, the OA and K values were higher than the Sentinel-2 at 98.6% (K = 97.4%), 98.5% (K = 97.3%), and 98.6% (K = 97.4%) for RF, SVM, and the ensemble, respectively. Also, the final combination of NDVI and HBSI saw a slight improvement in overall accuracy for training and validation when compared to the other sets of covariates (e.g., 6 VIs, 4 SIs, 3 VIs and 2 SIs) for both Sentinel-2 and AVIRIS-NG images.

**Table 3.** Tabulated accuracy statistics for RF, SVM, and their ensemble for the different sets of covariates. All *p*-values are less than 0.001. OA = overall accuracy (%), K = kappa (%).

| Variable Set | Sentinel-2 | | | | AVIRIS-NG | | | |
| --- | --- | --- | --- | --- | --- | --- | --- | --- |
| | Training | | Validation | | Training | | Validation | |
| | OA | K | OA | K | OA | K | OA | K |
| **6 VIs** | | | | | | | | |
| RF | 92.4 | 89.6 | 92.3 | 89.2 | 97.9 | 96.7 | 97.9 | 96.7 |
| SVM | 91.6 | 87.3 | 91.8 | 87.4 | 97.1 | 95.7 | 97.0 | 95.5 |
| Model Ensemble | 92.0 | 88.9 | 91.8 | 87.5 | 97.3 | 96.1 | 97.0 | 96.9 |
| **4 SIs** | | | | | | | | |
| RF | 90.6 | 89.7 | 90.5 | 89.9 | 98.2 | 97.2 | 98.5 | 97.1 |
| SVM | 91.4 | 88.6 | 91.9 | 87.7 | 97.1 | 96.2 | 97.3 | 96.2 |
| Model Ensemble | 90.3 | 87.9 | 90.5 | 87.2 | 97.7 | 97.0 | 98.1 | 97.1 |
| **3 VIs & 2 SIs** | | | | | | | | |
| RF | 91.2 | 87.9 | 90.3 | 88.1 | 98.2 | 97.3 | 98.7 | 97.2 |
| SVM | 91.5 | 88.2 | 92.4 | 87.7 | 95.7 | 94.6 | 95.7 | 93.5 |
| Model Ensemble | 91.4 | 88.0 | 91.9 | 87.1 | 96.8 | 95.9 | 96.7 | 95.2 |
| **NDVI & HBSI** | | | | | | | | |
| RF | 93.9 | 88.1 | 93.5 | 87.8 | 98.2 | 97.3 | 98.6 | 97.4 |
| SVM | 94.0 | 89.3 | 94.9 | 90.0 | 98.2 | 97.4 | 98.5 | 97.3 |
| Model Ensemble | 92.0 | 87.7 | 93.6 | 87.9 | 98.2 | 97.4 | 98.6 | 97.4 |

A matrix of class accuracies using the final model of NDVI and HBSI (Table 4) are shown in terms of the Kappa coefficient and overall accuracy. For Sentinel-2, the overall accuracies were 93.5%, 94.9%, and 93.6% for RF, SVM, and ensemble models, respectively. The kappa values range from 87.8 to 90.0, which indicate an acceptable level of accuracy of the classified maps. However, for the AVIRIS-NG, the overall accuracies were higher at 98.6%, 98.5%, and 98.6%, for RF, SVM, and ensemble models, respectively. The range of kappa values was also higher compared to Sentinel-2, with a range of 97.3 to 97.4.

**Table 4.** Confusion matrix from validation dataset using the final model of NDVI and HBSI.

| | Sentinel-2 | | | | | | | | |
|---|---|---|---|---|---|---|---|---|---|
| | **RF** | | | **SVM** | | | **Ensemble** | | |
| | **Soil** | **Urban** | **Others** | **Soil** | **Urban** | **Others** | **Soil** | **Urban** | **Others** |
| Soil | 1324 | 92 | 13 | 1393 | 78 | 33 | 1317 | 82 | 13 |
| Urban | 96 | 2947 | 2 | 88 | 2971 | 2 | 106 | 2957 | 2 |
| Others | 93 | 23 | 376 | 32 | 13 | 256 | 90 | 23 | 376 |
| Overall | | 93.5 | | | 94.9 | | | 93.6 | |
| Kappa | | 87.8 | | | 90.0 | | | 87.9 | |
| | **AVIRIS-NG** | | | | | | | | |
| Soil | 8744 | 45 | 173 | 8636 | 72 | 20 | 8744 | 47 | 175 |
| Urban | 150 | 2155 | 18 | 104 | 2080 | 5 | 150 | 2153 | 18 |
| Others | 3 | 1 | 17,237 | 157 | 49 | 17,403 | 3 | 1 | 17,235 |
| Overall | | 98.6 | | | 98.5 | | | 98.6 | |
| Kappa | | 97.4 | | | 97.3 | | | 97.4 | |

Table 5 lists the accuracy statistics after we ran the RF-SVM ensemble model using the individual soil indices. Similar to the results in Table 2, the HBSI and NDSI2 were the two indices with over 90% in overall accuracy, for both Sentinel-2 and AVIRIS-NG images. The BI had the lowest OA overall.

**Table 5.** Tabulated accuracy statistics for the individual soil index using an ensemble of RF and SVM models. All *p*-values are less than 0.001. OA = overall accuracy (%), K = kappa (%).

| | Sentinel-2 | | | | AVIRIS-NG | | | |
|---|---|---|---|---|---|---|---|---|
| **Soil Index** | **Training** | | **Validation** | | **Training** | | **Validation** | |
| | **OA** | **K** | **OA** | **K** | **OA** | **K** | **OA** | **K** |
| BSI | 89.4 | 88.5 | 89.2 | 88.8 | 92.3 | 90.7 | 92.1 | 90.9 |
| BI | 88.6 | 86.1 | 88.2 | 86.3 | 89.3 | 88.6 | 89.5 | 88.7 |
| NDSI2 | 90.0 | 88.4 | 90.1 | 87.7 | 93.7 | 92.3 | 92.9 | 91.8 |
| HBSI | 91.3 | 88.6 | 91.7 | 87.7 | 94.2 | 94.4 | 94.1 | 93.6 |

## 4. Discussion

### 4.1. Characteristics of HBSI vs. Other Indices

The results of our study indicate that the HBSI outperformed other existing indices for mapping bare soil. In our analysis, we observed that the HBSI, which is calculated using the blue, green, NIR, and SWIR2 spectral bands, exhibited unique advantages compared to other indices.

One distinctive characteristic of the HBSI is its utilization of the blue band in addition to the NIR and SWIR2 bands. These bands demonstrated the capability of the HBSI to discriminate bare-soil features from urban and other land-use classes. Unlike other bare-soil indices that only utilize SWIR2 and green bands [11,30], the HBSI took advantage of the reflectance of the blue band that allows for the capture of additional information. Furthermore, by combining the blue band with the NIR and then normalizing the difference, the dissimilarities between the strongest and weakest features within these spectral regions were emphasized. A similar study by Liu et al. [44] also highlighted the SWIR and the blue bands as being the two important spectral bands for bare-soil mapping since they represent the highest and the lowest reflectance for soil, respectively.

The HBSI is a promising alternative to existing bare-soil indices as it dramatically widens the gap between bare soil and other classes based on their unique spectral features. When used alone, the HBSI showed its effectiveness in bare-soil mapping with over 91% accuracy for Sentinel-2 and AVIRIS-NG images.

Finding the best combination of indices is an important step towards achieving the efficiency needed for mapping and understanding soil behavior. The combination of HBSI and NDVI showed a slightly better performance in bare-soil classification, with >92% for Sentinel-2 images and >97% for AVIRIS-NG images. The combination of indices minimized vegetation influence and maximized bare-soil features, which resulted in positive classification for bare-soil pixels only [15]. These VIs were used by several authors to classify bare soil, either separately or in combination [26–29]. A study in Italy using Sentinel-2 combined NDVI and BSI and delivered good discrimination between bare soil and other land classes [12].

### 4.2. Limitations of HBSI

While the HBSI has been proven useful for Sentinel-2 and AVIRIS-NG images in this study and it shows promise, it is important to acknowledge some limitations and uncertainties associated with the results. Since the HBSI uses the SWIR2 band (~2080–2350 nm), the potential of the index should be validated with other satellite images that contain SWIR2 and with broader spatial resolutions, such as Landsat 8/9. Second, since the spectral behavior of bare soil can vary with different soil types or landscapes [45], the HBSI could behave differently when applied to mudflats or dunes. We only tested the HBSI in a study area that is considered as farmland, with diverse agricultural management practices. More tests on other soil landscapes are needed, and caution should be exercised when applying the HBSI to different regions. Third, this study did not include a temporal analysis of bare soil. Seasonal soil mapping is needed to differentiate between bare lands that are fallow agriculture fields from those that are construction sites [15]. Fourth, because the HBSI relies on the SWIR2 wavelength, the index may not be useful for unmanned aerial vehicle (UAV) or drone images, for which SWIR is often unavailable. Lastly, although the combination of the HBSI and NDVI improved classification accuracy, there may still be cases where misclassifications occur, particularly in complex landscapes, or patches of urban areas (Figure 2c), or under challenging conditions. Future research could focus on addressing these limitations to further improve the accuracy and robustness of bare-soil mapping using the HBSI.

### 4.3. Performance of Ensemble Model

The effectiveness of an ensemble model depends on the precision of the individual models and conditional bias in simulated values during model training [46]. Machine learning techniques enable automated feature selection and extraction, identifying the most relevant spectral bands or indices for mapping bare soil. By focusing on informative features, these machine learning models could improve the accuracy and efficiency of the mapping process [40]. Combining machine learning models into a single ensemble produced a more representative soil map since it highlighted the agreement across algorithms [47]. In our study, the RF-SVM ensemble surpassed the performance of the individual models. In other words, the ensemble was even stronger as a result of the rather high performance of the individual models, despite the fact that prior research described the ensemble algorithm as having limited interpretability [48]. We may test additional machine learning models in the future, add ensemble criteria to exclude underperformers, and broaden the sorts of models chosen to boost ensemble performance.

### 4.4. HBSI Potential Area of Focus for Future Reseearch

There are several ways in which future research can improve the HBSI and its application in bare-soil mapping, such as the fusion of multi-temporal data. Bare soil conditions can vary over time due to changes in season and land management practices, and due to natural disturbances [15]. By integrating multi-temporal hyperspectral data, we will be able to capture temporal patterns and enhance the performance of the HBSI for mapping bare soil. Time series analysis, data-fusion techniques, and change-detection algorithms could be applied to utilize multi-temporal data and improve mapping accuracy. Second, the

integration of ancillary data [29], such as soil moisture measurements, terrain information, or soil-texture data, could also improve the accuracy and reliability of bare-soil mapping using the HBSI. Machine learning techniques could be utilized to effectively combine and exploit these additional data sources.

## 5. Conclusions

In this paper, we developed a bare-soil spectral index called the HBSI and investigated its effectiveness using two machine learning algorithms applied to two different satellite images. The statistical tests showed that the HBSI performs better when compared with other bare-soil indices and could meet the requirements for bare-soil classification. Mapping bare soil from Sentinel-2 (multispectral) and AVIRIS-NG (hyperspectral) images, as proposed in this study, is feasible and reliable. We believe in the potential of HBSI to improve the accuracy of bare-soil remote-sensing mapping.

**Author Contributions:** Conceptualization, E.A.L.S.; methodology, E.A.L.S.; software, S.S.K.; validation, E.A.L.S.; formal analysis, E.A.L.S.; investigation, E.A.L.S.; resources, S.S.K.; data curation, E.A.L.S.; writing—original draft preparation, E.A.L.S.; writing—review and editing, S.S.K.; visualization, E.A.L.S.; supervision, S.S.K.; project administration, S.S.K.; funding acquisition, S.S.K. All authors have read and agreed to the published version of the manuscript.

**Funding:** This work was supported by the National Aeronautics and Space Administration (Grant number 80NSSC17K0653 P00001) for the joint NASA and Indian Space Research Organization AVIRIS-NG Campaign in India. The study was also supported by NIFA/USDA through Central State University Evans-Allen Research Program Fund Number NI201445XXXXG018-0001.

**Data Availability Statement:** Not applicable.

**Conflicts of Interest:** The authors declare no conflict of interest.

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
