# Peer review of "Hyperspectral Bare Soil Index (HBSI): Mapping Soil Using an Ensemble of Spectral Indices in Machine Learning Environment"

_land, doi:10.3390/land12071375_

Round 1

Reviewer 1 Report

I suggest authors should add one sentence to clarify why the mapping of bare soil in a complex urban region is still a challenge. Moreover, please also add another sentence to answer can people can address this, after which you can say your intention to develop an index is a good way.

Line 21, please add one sentence to declare the novelty of this paragraph.

Line 33, please add at lease one sentence to justify why bare land was mostly misconducted.

Line 34, I am not sure if the proposal of a new index is the only way to address this question. Is it a technical issue or an index-related issue?

Please expand the information in line 49-55.

Please expand and add some references to line 56-58. Like: Predicting soil physical and chemical properties using vis-NIR in Australian cotton areas. Catena, 196, 104938.   Soil exchangeable cations estimation using Vis-NIR spectroscopy in different depths: Effects of multiple calibration models and spiking. Computers and Electronics in Agriculture, 182, 105990.   Clay content mapping and uncertainty estimation using weighted model averaging. Catena, 209, 105791.

Line 161, authors are suggested to justify why only RF and SVM were selected. They are actually not good in prediction. Please also tailor the description to your explanations.

Subsection 2.6, please add a paragraph to indicate which indicators you will use to assess the performance.

Authors have not presented a sound analysis of their results. The discussion section very simple as well. Please expand.

Please improve.

Reviewer 2 Report

Dear authors,

I have carefully reviewed your article titled "Hyperspectral Bare Soil Index (HBSI): Mapping soil using an ensemble of spectral indices in machine learning environment" and found it to be a valuable contribution to the field. However, I have a few recommendations to enhance the clarity and comprehensiveness of your work. Please find below my suggestions for revision:

Results and Discussion:

Further elaborate on the results in the discussion section to interpret them more comprehensively. For instance, discuss the potential reasons why the Hyperspectral Bare Soil Index (HBSI) outperformed other existing indices, providing insights into the unique advantages of the HBSI. Additionally, discuss any limitations or uncertainties associated with the results and suggest areas for future research.

Reviewer 3 Report

The paper mainly deals with the mapping soil using ensemble of spectral indices submitted to LAND. I have some fundamental issues that need to be clarified. These are as follows:

1. The soil mapping was carried out in this study but why vegetation indices have been employed? It needs to be clarified.

2. How the indices were calculated from SENTINEL data? Is this same as LANDSAT? I missed this issue. Better to clarify the issue.

3.  If the resolution of the images are different, then how it can be comparable [as mentioned in section 2.2]? Please clarify this.

4.  What are the significant of using machine learning?

5. How it has been validated? It is not clear.

6. Discussion section is very poor, it needs to be supported from the literatures.

The author can follow the following literatures and hope it can be helpful:

1.         Liu, Z., Xu, J., Liu, M., Yin, Z., Liu, X., Yin, L.,... Zheng, W. (2023). Remote sensing and geostatistics in urban water-resource monitoring: a review. Marine and Freshwater Research. doi: 10.1071/MF22167

2.         2. Li, R., Wu, X., Tian, H., Yu, N., & Wang, C. (2022). Hybrid Memetic Pretrained Factor Analysis-Based Deep Belief Networks for Transient Electromagnetic Inversion. IEEE Transactions on Geoscience and Remote Sensing, 60. doi: 10.1109/TGRS.2022.3208465

3.         Zhu, W., Chen, J., Sun, Q., Li, Z., Tan, W.,... Wei, Y. (2022). Reconstructing of High-Spatial-Resolution Three-Dimensional Electron Density by Ingesting SAR-Derived VTEC Into IRI Model. IEEE Geoscience and Remote Sensing Letters, 19. doi: 10.1109/LGRS.2022.3178242

.

Round 2

Reviewer 1 Report

Authors have well addressed all my concerns.

There are some informal description.

Author Response

We are deeply grateful for their invaluable insights and constructive feedback on the manuscript.

Thank you.

Reviewer 2 Report

The article has significantly improved with the proposed changes; therefore, I have nothing further to add.

Author Response

(The authors gave the same response as above.)

Reviewer 3 Report

The paper has been revised as suggested. Now, it can be accepeted for publications. 

Author Response

(The authors gave the same response as above.)
